# Infant Botulism: Checklist for Timely Clinical Diagnosis and New Possible Risk Factors Originated from a Case Report and Literature Review

**DOI:** 10.3390/toxins13120860

**Published:** 2021-12-02

**Authors:** Robertino Dilena, Mattia Pozzato, Lucia Baselli, Giovanna Chidini, Sergio Barbieri, Concetta Scalfaro, Guido Finazzi, Davide Lonati, Carlo Alessandro Locatelli, Alberto Cappellari, Fabrizio Anniballi

**Affiliations:** 1Unità di Neurofiopatologia, Fondazione IRCCS Ca’ Granda Ospedale Maggiore Policlinico, 20122 Milan, Italy; sergio.barbieri@policlinico.mi.it (S.B.); alberto.cappellari@policlinico.mi.it (A.C.); 2Neurology Unit & MS Centre, Fondazione IRCCS Ca’ Granda Ospedale Maggiore Policlinico, 20122 Milan, Italy; mattia.pozzato@unimi.it; 3Dino Ferrari Centre, Neuroscience Section, Department of Pathophysiology and Transplantation, University of Milan, 20122 Milan, Italy; 4Pediatric Unit, Fondazione IRCCS Ca’ Granda Ospedale Maggiore Policlinico, 20122 Milan, Italy; lucia.baselli@policlinico.mi.it; 5Pediatric Intensive Care Unit, Fondazione IRCCS Ca’ Granda Ospedale Maggiore Policlinico, 20122 Milan, Italy; giovanna.chinidi@policlinico.mi.it; 6National Reference Centre for Botulism, Nutrition and Veterinary Public Health, Department of Food Safety, Istituto Superiore di Sanità, 00161 Rome, Italy; concetta.scalfaro@iss.it (C.S.); fabrizio.anniballi@iss.it (F.A.); 7Department of Food Control, Istituto Zooprofilattico Sperimentale della Lombardia e dell’Emilia-Romagna, 25124 Brescia, Italy; guido.finazzi@izsler.it; 8Toxicology Unit, Laboratory of Clinical and Experimental Toxicology, and Poison Control Centre and National Toxicology Information Centre, Istituti Clinici Scientifici Maugeri IRCCS, 27100 Pavia, Italy; davide.lonati@icsmaugeri.it (D.L.); carlo.locatelli@icsmaugeri.it (C.A.L.)

**Keywords:** infant botulism, hypogammaglobulinemia, cytomegalovirus, diagnosis, risk factor, diagnostic criteria

## Abstract

Infant botulism is a rare and underdiagnosed disease caused by BoNT-producing clostridia that can temporarily colonize the intestinal lumen of infants less than one year of age. The diagnosis may be challenging because of its rareness, especially in patients showing atypical presentations or concomitant coinfections. In this paper, we report the first infant botulism case associated with Cytomegalovirus coinfection and transient hypogammaglobulinemia and discuss the meaning of these associations in terms of risk factors. Intending to help physicians perform the diagnosis, we also propose a practical clinical and diagnostic criteria checklist based on the revision of the literature.

## 1. Introduction

Botulism is an acute neuromuscular junction (NMJ) disorder caused by botulinum neurotoxins (BoNTs) that, after entering the blood stream by various routes, block nerve function (neuromuscular blockade) through inhibition of the excitatory neurotransmitter acetylcholine’s release from the presynaptic membrane of neuromuscular junctions in the somatic nervous system [1,2].

BoNTs are produced by anaerobic spore-forming bacteria belonging to the genus *Clostridium* and called BoNT-producing clostridia. *Clostridium botulinum*, and rarely *Clostridium baratii* and *Clostridium butyricum*, have been recognized as responsible for producing toxins involved in human botulism [3]. Recently, whole genome sequencing and molecular investigations revealed that some strains are capable of producing BoNT/B clustered as *Clostridium sporogenes*, which is notably resembled as a non-toxigenic strain related to proteolytic *C. botulinum* [3]. Generally, strains produce one type of BoNT; however, some can produce two or three toxins (or mosaic toxins) or produce one toxin and harbor a silent gene [4].

According to their immunological features, BoNTs have been classified into 9 toxinotypes (A, B, C, D, E, F, G, H, or H/A, or F/A, X) and 41 subtypes [5,6]. In addition, some botulinum-toxin-like proteins have been identified from gene sequences of non-clostridial species, such as *Weissella oryzae*, *Chryseobacterium piperi*, and *Enterococcus faecium* [7,8,9,10,11,12].

Botulism is classified according to the neurotoxin production site and entering route. Foodborne botulism results after the consumption of foods or beverages contaminated with pre-formed neurotoxins. Iatrogenic botulism originates from improper use of the BoNTs for cosmetic/therapeutic purposes. Inhalation botulism results from the accidental/deliberate release of aerosolized toxin [13]. Wound botulism is due to the *Clostridium botulinum* outgrowth and toxinogenesis in a wound. Intestinal toxemia botulism occurs if BoNT-producing clostridia spores encounter favorable conditions to germinate, produce toxins, and temporarily colonize the large intestinal lumen of infants and adults. When the affected people are infants under one year of age, the disease is called infant botulism (IB). Conversely, if the involved patients are children over one year or adults, the illness is referred to as adult intestinal colonization botulism [2,13,14].

In infants, susceptibility to intestinal colonization of BoNT-producing clostridia is primarily associated with the perturbations and/or the immaturity of the gut microbiota, which is subjected to profound variations during the first year of life and the lack of *Clostridium*-inhibiting bile acids [15,16]. Studies on animal models confirmed that infant mice are naturally susceptible to *C. botulinum* colonization, showing a peak of vulnerability between 8 and 11 days of age, in a pattern similar to the peaking of susceptibility between 2 and 4 months of age in humans. Conversely, the intestinal microbiota of adult animals can prevent *C. botulinum* colonization [17,18,19]. Although the minimum infective dose of *C. botulinum* spores for infants is not known, studies from exposure to spore-containing honey allowed the estimation that 10–100 spores may be sufficient to trigger the colonization [19].

The maturation of the infant gut microbiota is a complex and continuous process involving phylogenetic and functional features. Diet, drug, hormonal and metabolic status, and the immune system contribute to shaping the individual gut microbiota. The effects of external factors, such as birth mode, antibiotic therapies, and diet, are relatively well studied. On the other hand, the mechanisms involved in the infant gut regulation (e.g., secretion of mucus glycans, immunoglobulin A, and bile acids) and in the microbial community dynamics (e.g., bacterial growth or replication rates, the interaction between the community members) remains poorly understood [20].

Although the diet is one of the most important factors influencing the composition of infant gut microbiota, there is not a clear cause–effect relationship between feeding (e.g., breast-feeding, formula-feeding, consumption of solid foods) and IB [15,21]. Formula-fed infants develop the disease at a significantly younger age with a more rapid and severe course than breast-fed cases [15,22]. In their elegant paper, Panditrao and colleagues assume that breastfeeding may indirectly prevent or delay colonization because human milk contains oligosaccharides that influence the gut microbiota composition. In addition, human milk includes immune factors, such as secretory IgA and lactoferrin, and substances that block bacterial mucosal attachment [23]. The differences in the gut microbial composition between breastfed and formula-fed infants are well established. Breastfed infants have a higher taxon from the protective bacterial belonging to Actinobacteria, whilst formula-fed infants have a higher level of proinflammatory bacterial class γ-Proteobacteria [24]. In addition, human milk oligosaccharides support the competitive growth of beneficial bacteria, such as *Bifidobacterium* spp. Conversely, formula feeding leads to a gut microbiota with an increased abundance of *Clostridium* species and *Enterobacteriaceae* species and a mostly proteolytic gut metabolism [25].

From 1976 to date, several research groups have studied the possible risk factors for this illness [21,23,26]. Besides gut microbiota composition/perturbation, slow intestinal motility (less than one bowel movement per day) was recognized as a predisposing factor. For infants younger than two months of age, cesarean delivery, birth order >1, and living in a windy area are significant risk factors. For infants older than two months of age, the risks factors consist of breastfeeding at onset, feeding cereal/sterile solids during the incubation period, and dust exposure in the home [23]. Dust exposure is probably the most significant risk factor for IB since BoNT-producing Clostridia are soil-dwelling and dust-borne anaerobic bacteria. Infants can be exposed to dust thought several routes: home dust, home renovation works, construction sites near where they live, and dust carried by parents (from their working site to home) [27,28].

Bowel abnormalities (e.g., Meckel’s diverticulum and intussusceptions), *Clostridium difficile*-associated colitis, and mucosa alterations induced by enteroviruses were also described as risk factors [27,29,30]. In addition, thiamine deficiency due to thiaminase I activity in the infant feces has been described as a complicating factor of this disease [31].

Ingestion of honey, corn syrup, and herb infusion is popularly known as the vehicle of *C. botulinum* spores and represents the most important food-related risk factors for IB [26,32,33]. Although infant foods and in particular milk powder do not represent a risk because they are produced under high-level safety, a possible link to milk powder was postulated [34]. Other potential foods and environmental sources of BoNT-producing Clostridia spores, such as powdered infant rice cereals and untreated well-water, have been identified; however, in almost all IB cases, the actual origin of spores remains unidentified [35,36,37].

As extensively reviewed elsewhere, the illness typically presents with constipation, generalized hypotonia, and bulbar palsies that require hospitalization [15,38]. Treatment is based on supportive care, often needing intensive care with mechanical ventilation and enteral feeding. Antibiotic therapy is not recommended and should be reserved for treatment of secondary infections (e.g., pulmonary or urinary tract infections), avoiding aminoglycoside and clostridiocidal drugs that can increase the intestinal BoNTs level and worsen the neurological symptomatology [39,40,41]. Recently, the California Department of Public Health scientists unsuccessfully carried out a sizeable antimicrobial susceptibility study to identify an antibiotic (to which BoNT-producing Clostridia were resistant) to safely treat IB patients [41]. On the other hand, Mazuet and colleagues reported a β-lactam resistance in *C. botulinum* type A, which is notoriously susceptible to those molecules [42]. Specific treatment consists of the injection of botulinum antitoxins. Although the equine antitoxin has been successfully used for patients’ treatment, the California Department of Health Services produces and worldwide distributes the Botulism Immune Globulin Intravenous (Human) (BIG-IV), a human-derived antitoxin capable of neutralizing type A and B toxins [43,44]. This latter was created to overcome the serum sickness, anaphylaxis, and potential sensitization to animal proteins due to the use of equine antitoxin [45]. In addition, BIG-IV has a half-life of approximately 28 days (compared with 5–7 days of equine antitoxin) and the capacity to neutralize types A and B toxins for at least six months [46]. Antitoxin administration significantly reduces the length of hospitalization and allows a safer antibiotic therapy.

In the present paper, we report the first IB case associated with Cytomegalovirus (CMV) coinfection and transient hypogammaglobulinemia and discuss the meaning of these associations in terms of risk factors. In addition, we propose a practical clinical and neurophysiological diagnostic criteria checklist based on the review of the literature, because the illness remains an underdiagnosed condition with insidious presentation, which need to be distinguished by their mimickers.

## 2. Case Presentation

A 6-month-old male infant was admitted to the emergency room because of fever, feeding/sucking difficulties, and sleepiness. He was born at term after an eutocic delivery. His birth weight was 3530 g. The patient was exclusively breastfed and had no history of honey or herb infusion consumption before the symptomatology onset, but her father was a carpenter working with home renovation. The abnormal movement behaviors in the first day raised the suspicion of seizures, but the video electroencephalography excluded this hypothesis, showing normal findings. On the second day, the infant became weak, showing an expressionless face, poor cry, constipation, and oxygen desaturation, requiring intubation and invasive mechanical ventilation. Brain magnetic resonance imaging (MRI) and cerebrospinal fluid (CSF) analysis did not show any abnormalities. In contrast, blood tests showed hypogammaglobulinemia [IgG 68 mg/dL (180–800), IgM 14 mg/dL (20–100), IgA < 5 mg/dL (6–60)] and urine analysis revealed high levels of CMV DNA. In light of this latter finding, the diagnosis of CMV infection was established. Congenital CMV was excluded by analyzing CMV DNA on a Guthrie card sampled at the neonatal age. Nerve conductions studies (NCSs) showed peripheral motor nerve impairment with reduced compound motor action potentials (CMAPs) and typical results of sensory nerve action potentials (SNAPs). Standard repetitive high-frequency nerve stimulation (RNS) was regular. Needle electromyography (EMG) showed denervation potentials and small motor unit action potentials (MUAPs) with reduced recruitment. Due to the suspicion of a motor GBS variant triggered by CMV, ganciclovir (5 mg/kg IV twice daily for 3 weeks) and intravenous immunoglobulin (IVIg, 0.4 g/kg IV for five days) were started. Despite this therapy, the patient’s clinical conditions worsened.

One week after, in the context of a severe respiratory failure and general paralysis prominently involving the cranial and proximal segments with relative deep tendon reflex sparing (reduced but still present), areflexic dilated pupils were noted and a new spinal tap confirmed the absence of albumin-cytologic dissociation. At the NCS, marked reduced CMAPs were identified with preserved normal sensory nerve action potentials. Right tibial nerve high-frequency (50 Hz) RNS (Figure 1) with a standard 10-stimuli train showed an incremental pattern of 50% at the 10th stimulus (not significant but with a recruiting trend). A prolonged RNS train with 99 stimuli (almost 2 s duration) was performed, showing a CMAP increment of around 100% (suggestive of pre-synaptic dysfunction). Needle EMG revealed small and short MUAPs, resembling the brief duration and small-amplitude overly abundant motor-unit action potentials (BSAPs) described in botulism together with poor recruitment.

IB was suspected, and after discussing the case with the Italian National Reference Center for Botulism and with the Pavia Poison Centre, rectal swab and rectal irrigation (enema) samples were collected and sent to the regional reference laboratory at the Istituto Zooprofilattico Sperimentale in Brescia for confirmation. HBAT equine antitoxin (10 mL/kg of body weight) was promptly administered followed by metronidazole 7.5 mg/kg IV every six hours for one day. Laboratory investigations confirmed the clinical suspicion through the recovery of BoNTs in enema (using the mouse bioassay and neutralization test) and simultaneously detection of the gene encoding for BoNT/B by real-time PCR. Proteolytic *C. botulinum* type B was isolated by all the specimens. Within a week from the antitoxin administration, the patient started to recover pupilar reflexes and distal limb movements. After 14 days, he was extubated and admitted to an intensive rehabilitation center. Within two months from disease onset, swallowing function returned to normal. He reached complete neurological recovery within 4 months (flowchart of the case is available in the Appendix A).

The hypogammaglobulinemia was extensively investigated. Beyond low IgA, IgG, and IgM levels, blood samples pre-IVIg showed a normal white blood cell count with normal lymphocyte subpopulations and an adequate antibody response to hepatitis B vaccine. Lymphocyte extended immunophenotyping displayed a normal distribution of all lymphocyte subsets. An extensive gene panel for agammaglobulinemia/hypogammaglobulinemia (with more than 50 genes) did not identify pathological genetic alterations in the coding regions. Subsequent blood evaluations demonstrated a gradual increase of serum immunoglobulins that reached the normal range after five months.

## 3. Results and Discussion

To the best of our knowledge, we report the first case of infant botulism associated with a CMV coinfection and transient hypogammaglobulinemia.

Because of the recovery of CMV DNA in the patient’s urine, the first diagnostic suspicion was Guillain–Barré syndrome (GBS), triggered by this virus. This first diagnostic hypothesis was highly suggestive because GBS presents a clinical picture similar to that of botulism. GBS is also included in differential diagnosis along with other etiologies, such as spinal muscular atrophy (SMA) type 1, Miller Fisher variant of GBS, Lambert–Eaton syndrome secondary to neuroblastoma, myasthenia gravis, brain stem encephalitis, meningitis, and metabolic disorders [47,48,49]. Although suspected sepsis and meningitis are the most common diagnoses at hospital admission, Khouri and colleagues published data on 76 clinical mimics identified in patients treated with BIG-IV from 2005 to 2015. According to this retrospective survey, the most common diagnosis that mimicked infant botulism remains SMA type 1 (15 patients), followed by mitochondrial disorders (4 patients), GBS or its variants (3 patients), and Parechovirus encephalitis (3 patients). Other pathologies were recognized in 19 patients whilst 32 infants remained without a definitive diagnosis or with a diagnosis of probable botulism lacking the laboratory confirmation [49]. Although the diagnosis of infant botulism is mainly based on clinical suspicion and the recovery of BoNTs or BoNT-producing Clostridia in clinical specimens is crucial for definitive diagnosis, electrodiagnostic studies can be helpful for timely diagnosis while waiting for the laboratory results [50,51,52]. However, the special ENMG testing needed for the specific diagnosis is complex, not available in all hospitals, and its sensitivity may vary. When an infant fulfils the clinical suspicion criteria for botulism and other more common differential diagnoses are ruled out by instrumental tests (no albumin-cytological dissociation at the cerebrospinal fluid analysis and/or normal contrast magnetic resonance imaging), the following ENMG findings allow localization of the disease site into the neuromuscular system: (i) low compound muscle action potential amplitudes; (ii) normal sensitive action potentials; and (iii) brief-duration and small-amplitude overly abundant motor-unit action potentials (termed as BSAPs) that remember the myogenic pattern and denervation (due to functional denervation of muscle fibers) [52]. In this clinical setting, microbiological investigations for botulism should be immediately requested and antitoxin promptly administered, although this neurophysiological pattern in itself is not specific only for IB.

The specific ENMG tests confirming the presynaptic block typical of IB and distinguishing it from other neuromuscular disorders are repetitive nerve stimulation (RNS) at high-rate and stimulated single-fiber electromyography (sSFEMG). Still, unfortunately, their sensitivity is variable [52]. In patients with botulism, high-rate RNS (>20 Hz) given for seconds (tetanic facilitation) or post-tetanic facilitation post-exercise single supramaximal stimulations (SSSs) may be able to produce the typical > 120% incremental increase in the CMAP amplitude compared to the basal CMAP [50,51,53]. The main problem of the specific ENMG tests is that the sensitivity for detecting the presynaptic disorder is variable, depending on the specific protocol applied, the patient age, stimulation site, timing, and severity of disease [54,55], so the absence of an incremental response to the nerve RNS stimulation cannot be considered a reliable test to exclude a diagnosis of infantile botulism. In non-collaborating patients, such as infants or paralyzed individuals, the voluntary SSSs is inapplicable; therefore, a tetanic stimulation test at 50 Hz for 10 s is requested to demonstrate the pathological incremental response of presynaptic neuromuscular junction disorders [50,56]. A conventional 10-stimulus train is often not able to reach a significant increment threshold, as exemplified by our case (Figure 1), and a prolonged stimulation may be needed. Because the test is painful, sedoanalgesia should be performed. An alternative to the tetanic test is the sSFEMG, which in IB cases may show an increased jitter at 2 Hz that reduces with higher rates of stimulation, beginning at 10 Hz and peaking at 20 Hz. On the contrary, the jitter increases with higher rates in patients affected by myasthenia gravis [54].

Although the magnetic resonance imaging (MRI) in IB patients generally shows a normal picture and can be used for revealing alternative causes of acute flaccid paralysis in infants, including demyelinating diseases, metabolic disorders, ischemia, spinal cord compression, and neoplastic processes, IB cases showing enhancement of the cervical nerve roots correlated with the bulbar weakness have been reported [57].

In the case described in the present paper, CMV might have played a role in delaying the diagnosis and, as a risk factor for IB, tampering with gut microbiota and host immunity. Santos-Rocha and colleagues demonstrated in animal models that CMV enhanced host colonization by butyrate-producing bacteria [58]. Among these organisms, the species belonging to the genus *Clostridium* that significantly increased their abundance were some members of *Clostridium* cluster IV and XIVa [58]. Although BoNT-producing clostridia are included in cluster I and may not be affected by this phenomenon, Shirey and colleagues found that they were present in a meagre amount in the feces of infants affected by IB, indicating that the disease can be caused by a relatively limited number of these organisms [59].

In addition to CMV infection, the hypogammaglobulinemia may also have affected the gut microbiota imbalance by favoring the *C. botulinum* colonization. IgA plays a crucial role in inhibiting pathogen colonization and proliferation, and neutralizing bacterial toxins and enzymes. Secretory IgA (SIgA) is also decisive in maintaining stable intestinal microbiota composition, promoting biofilm formation and colonization by commensal organisms, and regulating intestinal homeostasis [60]. In their exciting paper, Matsumura and colleagues reported that SIgA can capture type B botulinum toxins and attenuate the binding activity of the toxin to intestinal epithelial cells [61].

Although further studies are needed to confirm the role of CMV coinfection and hypogammaglobulinemia as possible risk factors for IB, we suggest investigating IgA, IgG, and IgM levels in the blood of patients with suspected botulism and CMV DNA in their specimens.

The two significant limitations of this paper are the lack of CMV detection in the patient’s stool and the inability to perform the intestinal microbiota characterization through metagenomics. These two laboratory investigations would have allowed us to understand better the relationship between CVM and IgA deficiency and *C. botulinum* colonization.

On a worldwide basis, IB is a rare disease. The United States reported the most significant number of cases, notifying 80–100 cases annually. In 2019, for example, a total of 152 IB cases were notified by the CDC. The largest incidence rate (6.5 cases per 100,000 live births) is reported by California, which notified 1345 laboratory-confirmed cases in 1976–2016 [62]. In Canada, from 1979 to 2019, 63 cases were laboratory-confirmed [63]. Among European countries, Italy reported the largest number of laboratory-confirmed cases and, from 1984 to October 2021, notified 55 cases. Considering its rareness, several physicians do not see IB cases during their medical activity, and the diagnosis could represent a challenge. To overcome these difficulties, the evidence-based tool we propose in Table 1 can help physicians with low familiarity diagnose and address IB in the correct management of the disease.

## 4. Conclusions

IB can cause a broad spectrum of presentations, from mild hypotonia to fatal outcomes. Since literature reported an increased number of clinical mimickers and/or atypical pictures that can hamper the diagnosis, front-line awareness remains crucial for posing clinical suspicion, timely diagnosis, and proper management of the disease. Physicians should suspect IB in hypotonic infants presenting with feeding/sucking difficulties, ptosis, and constipation, and also in patients without a history of honey consumption. In patients showing clinical pictures that are not totally consistent with IB, other diagnostic hypotheses should be considered, and differential diagnosis promptly started. Positive laboratory results are essential for a definitive diagnosis; however, laboratory investigations are time-consuming and there may be a wait of several days to obtain results. Pending laboratory results, electrodiagnosis could help exclude alternate diagnoses and move towards the correct management of the patient.

## 5. Materials and Methods

### 5.1. Diagnosis

Observation of clinical signs at the hospital admission and clinical laboratory analysis oriented the first diagnostic hypothesis. Respiratory and general paralysis progression and electrophysiologic tests triggered the clinical suspicion of botulism, which was confirmed by laboratory investigations, as reported elsewhere [64].

### 5.2. Literature Review on Clinical and Instrumental Criteria for Early Diagnosis

A PubMed search for relevant published articles on clinical and instrumental criteria for early diagnosis was performed by combining the following keywords: “infant botulism, diagnosis, early diagnosis, differential diagnosis, electrodiagnosis, electromyography, nerve impulse, neurologic examination, evoked potentials, electrophysiologic testing, repetitive nerve stimulation, single-fiber electromyography, nerve conduction studies, compound muscle action potentials”.

A second PubMed search was performed by combining the following keywords: “infant botulism, risk factors, predisposing factors, coinfection, virus, virus coinfections, cytomegalovirus, hypogammaglobulinemia”. The searches were set up from 1976 to September 2021, considering only papers published in English.

The references of the retrieved articles were used to identify other relevant studies. The literature revisions were carried out by two independent authors using a three-step process in which the appropriateness of the title (first step), abstract (second step), and full paper (third step) were assessed.

## Figures and Tables

**Figure 1 toxins-13-00860-f001:**
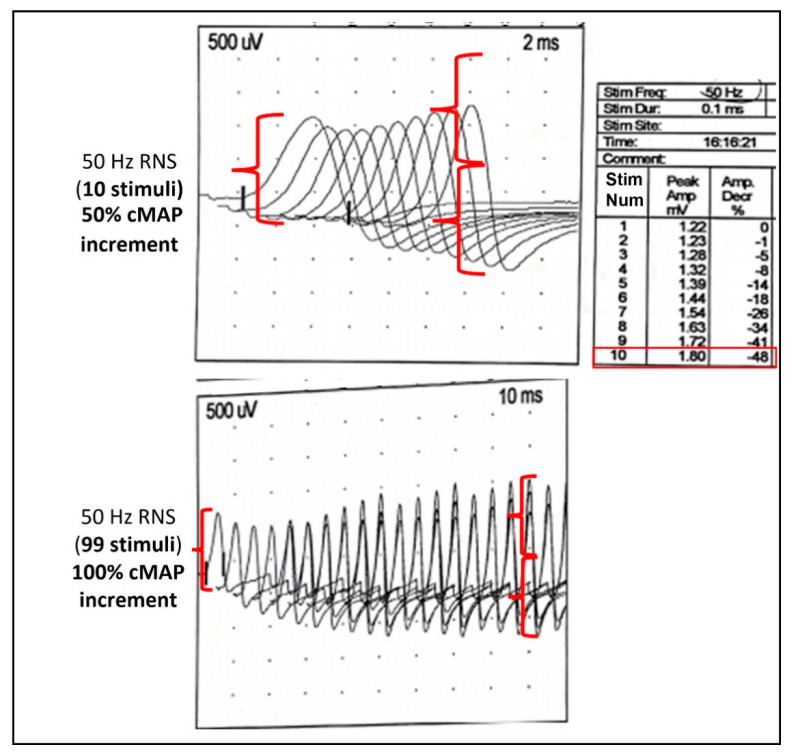
Repetitive nerve stimulation (RNS) at high frequency (50 Hz) of the right tibial nerve from the abductor hallucis muscle. cMAP, compound motor action potential.

**Table 1 toxins-13-00860-t001:** Proposed clinical criteria for IB diagnosis.

Infant Botulism Clinical and Instrumental Suspicion Checklist	
	**A. Clinical criteria**
A1	Acute weakness with descending progression as initial cranial involvement as poor feeding, poor suck, fatigability when eating, expressionless face, drooling, weakness or change of cry, lethargy, respiratory insufficiency.
A2	Autonomic signs as altered pupillary reflexes, constipation.
A3	Deep tendon reflex (DTR) normal or reduced, often with relative sparing in comparison with the paralysis degree *
A4	Presence of environmental risk factors as contact with soil (living in rural areas or home renovation environments, parental work in contact with soil and home renovation, honey, or herbal tea ingestion, contact with botulism outbreaks).
A5	Presence of personal predisposing factors as clinical conditions influencing intestinal microbiota and immunity (e.g., recent viral infection history, etc.)
	**B. Brain and spinal MRI**
B1	No parenchymal lesions explaining signs.
B2	No cranial and spinal root MRI gadolinium enhancement (a sign of nerve root inflammation usually found in GBS, the most important mimicker).
	**C. Lumbar puncture**
C1	Normal CSF parameters.
C2	No albumin-cytologic dissociation (an inflammation sign usually found in GBS, the most important mimicker).
	**D. Electrodiagnosis**
	**ENMG basic criteria compatible with IB**
D1	M-NCS: Low CMAP amplitudes.
D2	S-NCS: Normal SNAP amplitude.
D3	N-EMG: Brief-duration and small-amplitude, overly abundant motor-unit action potentials (termed as BSAPs) similar to a myopathic pattern, possible denervation potentials.
	**ENMG specific criteria proving the pre-synaptic block (pathophysiology of IB) ****
D4	RNS: Tetanic stimulation at 50 Hz for 10 s to prove incremental response of cMAP in comparison with a basal reduced CMAP. Being a potential painful test, use sedoanalgesia.
D5	S-SFEMG: increased jitter indicating a NMJ disorder corrected at higher frequency stimulation.
	**E. Laboratory studies usually on clinical specimens (definitive confirmation) *****
E1	Detection of BoNT-producing Clostridia from fecal specimens.
E2	Detection of BoNTs from fecal specimens (and serum).

A. Clinical criteria for IB suspicion criteria: if A1 and A2 are present in a patient < 1 year, IB should be always rapidly considered. Presence of one or more among A3, A4, A5 reinforce suspicion. When A1 and A2 are present and there is no prove of alternative diagnosis causing an acute flaccid paralysis (see C and/or D), a basic ENMG should be performed to find evidence of motor neuromuscular involvement (D1, D2, D3) and proceed to laboratory studies (E1, E2) and treatment with botulinum anti-toxin. Whenever possible, advances studies to prove the specific pathophysiology of IB and identifying the pattern of pre-synaptic block should be carried out to perform diagnosis (D4, D5). ENMG, electroneuromyography; MRI, Magnetic Resonance Imaging; CSF, cerebrospinal fluid; M-NCS, motor nerve conduction studies; S-NCS, sensory nerve conduction studies; N-EMG, needle electromyography; RNS, repetitive nerve stimulation; S-SFEMG, stimulated single fiber EMG; PCR, Polymerase chain reaction. *DTR sparing is not mandatory, in very severe flaccid they can be absent. ** These special techniques may be complex and are not available everywhere so should not block stop indication to microbiological tests. *** Clinical samples suitable for laboratory diagnosis purposes are faces (or enema and rectal swab if the patient is constipated). Blood serum can be tested for toxin; however, in IB cases, there is a low level of circulating toxin in the blood stream and high level of toxins in feces.

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
