# Peer review of "Infant Botulism: Checklist for Timely Clinical Diagnosis and New Possible Risk Factors Originated from a Case Report and Literature Review"

_toxins, 2021, doi:10.3390/toxins13120860_

Round 1
Reviewer 1 Report
Dear Editor,
this article is an interesting case report of infant botulism in association with CMV and transient hypogammaglobulinemia including a review of the literature and suggesting a workup-checklist as a help for clinicians to diagnose infant botulism. The authors present interesting aspects of the disease and important differential diagnostic considerations. The proposed clear checklist might help clinicians to diagnose infant botulism and manage it early.
I have only a few comments the authors might want to consider:
1. As this article is mainly a case report with review of literature, these words should be considered to appear in the headline of this case report to inform the reader.
2.
Line 8: "In this study,......"
Line 271: ...."of this study".....
The authors may want to rephrase the word "study" in these sentences as the authors describe a case report. The word study might confuse the reader in this context .
3.
Line 267-270: "Although further studies are needed to confirm the role of CMV coinfection and hypogammaglobulinemia as possible risk factors for IB, we suggest investigating IgA, IgG, and IgM levels in the blood of patients with suspected botulism and CMV DNA in their specimens."
The authors might want to comment, why they do not believe that the CMV infection and transient hypoglobulinemia are a conincidence in this case of IB as they suggest to look for CMV and hypoglobulinemia in suspected botulism.
Author Response
We thank Reviewer 1 for the time spent amelioration our work. We appreciate this effort.
I have only a few comments the authors might want to consider:
- As this article is mainly a case report with review of literature, these words should be considered to appear in the headline of this case report to inform the reader.
We modified the headline of the manuscript as “Infant botulism: checklist for timely clinical diagnosis and new possible risk factors originated a case report and literature review”.
2.
Line 8: "In this study,......"
Line 271: ...."of this study".....
The authors may want to rephrase the word "study" in these sentences as the authors describe a case report. The word study might confuse the reader in this context .
Thanks for the comment. We rephased the test accordingly.
Line 267-270: "Although further studies are needed to confirm the role of CMV coinfection and hypogammaglobulinemia as possible risk factors for IB, we suggest investigating IgA, IgG, and IgM levels in the blood of patients with suspected botulism and CMV DNA in their specimens."
The authors might want to comment, why they do not believe that the CMV infection and transient hypoglobulinemia are a coincidence in this case of IB as they suggest to look for CMV and hypogammaglobulinemia in suspected botulism.
Thank you for the comment. Literature reported that CMV and hypogammaglobulinemia may affect gut imbalance. We discuss this topic from line 254 to line 281. Since gut imbalance is the most important risk factor favouring the C. botulinum colonization in infants, we suggest to physicians investigating these parameters. New evidence on the concomitant CMV infections or hypogammaglobulinemia in infants presenting botulism colonization can reinforce our insights, improving knowledge and understanding on the pathophysiology of this rare but severe disease. To avoid redundances in the text we decided not include new sentences in the manuscript.
In addition we performed the following changes:
- In case presentation: a correction in Figure 1, as there was a mistake in the comment: with 99 stimuli high frequency stimulation in our patient the increment was 100% (as 200% was the value of CMAP compared to the basal value).
- In discussion: addition of a sentence in line 240: “so the absence of an incremental response to the nerve RNS testing cannot be considered a reliable test to exclude a diagnosis of infantile botulism”.
Reviewer 2 Report
General comments:
The topic is of interest. The present work draws attention to the case presentation of an infant with botulism associated cytomegalovirus coinfection and transient hypogammaglobulinemia. Furthermore, was discussed the meaning of these associations in terms of possible risk factors. A benefit of this paper is the proposal of a practical clinical and diagnostic criteria checklist.
General comments:
Do the authors consider a congenital CMV? Unfortunately, this aspect is not covered in the paper.
Reason for a severe respiratory (L 156). Respiratory tract infection due to transient hypogammaglobulinemia (THI) in the infant (?)
The authors are advised to present a flow chart of the case clinical presentation.
Comments in the main test:
L26 – L29: reference should be included
L40: …the consumption of foods or beverages
L286: Clinical criteria for IB suspicion – proposed Clinical criteria for IB diagnosis (as is written in the manuscript)
Author Response
The topic is of interest. The present work draws attention to the case presentation of an infant with botulism associated cytomegalovirus coinfection and transient hypogammaglobulinemia. Furthermore, was discussed the meaning of these associations in terms of possible risk factors. A benefit of this paper is the proposal of a practical clinical and diagnostic criteria checklist.
We thank Reviewer 2 for the time spent amelioration our work.
General comments:
Do the authors consider a congenital CMV? Unfortunately, this aspect is not covered in the paper.
Thank you for this comment. Congenital CMV was excluded by analysing CMV DNA on Guthrie cards sampled in the neonatal age. We included this sentence in the manuscript.
Reason for a severe respiratory (L 156).
Thanks for the comment. In the text, there was a mistake, and the word “failure” was missed. We modified the text accordingly.
Respiratory tract infection due to transient hypogammaglobulinemia (THI) in the infant (?)
The patient did not have a respiratory infection.
The authors are advised to present a flow chart of the case clinical presentation.
The flow chart was done and included as an additional file.
Comments in the main test:
L26 – L29: reference should be included
The reference was included as requested.
L40: …the consumption of foods or beverages
The word “beverages” was included in the text as requested.
L286: Clinical criteria for IB suspicion – proposed Clinical criteria for IB diagnosis (as is written in the manuscript)
The title of Table 1 was modified as suggested.
In addition we performed the following changes:
- In case presentation: a correction in Figure 1, as there was a mistake in the comment: with 99 stimuli high frequency stimulation in our patient the increment was 100% (as 200% was the value of CMAP compared to the basal value).
- In discussion: addition of a sentence in line 240: “so the absence of an incremental response to the nerve RNS testing cannot be considered a reliable test to exclude a diagnosis of infantile botulism”.